# A Nanoparticle’s Journey to the Tumor: Strategies to Overcome First-Pass Metabolism and Their Limitations

**DOI:** 10.3390/cancers14071741

**Published:** 2022-03-29

**Authors:** Joshua J. Milligan, Soumen Saha

**Affiliations:** Department of Biomedical Engineering, Duke University, Durham, NC 277018, USA; joshua.milligan@duke.edu

**Keywords:** solid tumor, nanomedicine, first-pass metabolism, RES blockade

## Abstract

**Simple Summary:**

Traditional cancer therapeutics suffer from off-target toxicity, limiting their effective dose and preventing patients’ tumors from being sufficiently treated by chemotherapeutics alone. Nanomedicine is an emerging class of therapeutics in which a drug is packaged into a nanoparticle that promotes uptake of the drug at a tumor site, shielding it from uptake by peripheral organs and enabling the safe delivery of chemotherapeutics that have poor aqueous solubility, short plasma half-life, narrow therapeutic window, and toxic side effects. Despite the advantages of nanomedicines for cancer, there remains significant challenges to improve uptake at the tumor and prevent premature clearance from the body. In this review, we summarize the effects of first-pass metabolism on a nanoparticle’s journey to a tumor and outline future steps that we believe will improve the efficacy of cancer nanomedicines.

**Abstract:**

Nanomedicines represent the cutting edge of today’s cancer therapeutics. Seminal research decades ago has begun to pay dividends in the clinic, allowing for the delivery of cancer drugs with enhanced systemic circulation while also minimizing off-target toxicity. Despite the advantages of delivering cancer drugs using nanoparticles, micelles, or other nanostructures, only a small fraction of the injected dose reaches the tumor, creating a narrow therapeutic window for an otherwise potent drug. First-pass metabolism of nanoparticles by the reticuloendothelial system (RES) has been identified as a major culprit for the depletion of nanoparticles in circulation before they reach the tumor site. To overcome this, new strategies, materials, and functionalization with stealth polymers have been developed to improve nanoparticle circulation and uptake at the tumor site. This review summarizes the strategies undertaken to evade RES uptake of nanomedicines and improve the passive and active targeting of nanoparticle drugs to solid tumors. We also outline the limitations of current strategies and the future directions we believe will be explored to yield significant benefits to patients and make nanomedicine a promising treatment modality for cancer.

## 1. Introduction

Research conducted in recent years has established nanomedicine as a promising tool to revitalize chemotherapeutics as first-line cancer treatments by improving their accumulation to solid tumors and enhancing their overall therapeutic window [1,2,3,4,5,6]. Since the approval of liposomal doxorubicin—marketed as Doxil—in 1995, nanomedicine for chemotherapy has seen unprecedented traction. A wide variety of nanomaterials including, but not limited to, liposomes [7,8,9,10,11], metallic nanoparticles (NPs) [12,13], solid lipid nanoparticles [14,15], polypeptides and other micelles [16,17,18,19], dendrimers, and other polymeric nanoparticles [20,21,22] have been tested at the bench in a hope to find a “magic bullet” for cancer treatment. Despite this, innovation in the field continues to fall short of adding a significant hit to the portfolio of clinically approved nano-drug formulations that translated to the clinic. Strikingly, as of February 2022, there is only one active cancer nanomedicine clinical trial in the United States which is in stage-IV—focusing only on liposomal formulations—as per the National Institute of Health (NIH) database (Table 1). Although nanomedicine was able to prolong the circulation time and mitigate some of the toxicities of free drugs, the improvement of overall therapeutic benefit is not significant in multiple clinical trials [23,24]. Rapid clearance of circulating NPs by first-pass metabolism has been identified as a major contributor for such dismal performance [25,26]. The first-pass effect is a process by which a drug gets metabolized before reaching its site of action, resulting in a reduced concentration of the active drug in target tissue. The first-pass effect is often associated with the liver, as this is a major site of drug metabolism. However, the first-pass effect can also take place in other metabolically active organs [27,28]. The extent of this effect varies patient-to-patient [29]. It is extremely important to consider the first-pass effect while determining the dosing of nanomedicines [28]. Further complicating this picture is the opsonization of various serum proteins onto nanoparticles upon *i.v.* administration, creating a protein corona with an associated “biological identity” that alters the fate of the particle in circulation [13,30,31,32]. Many of these proteins—known as opsonins—tag the nanoparticle for rapid degradation through phagocytosis by the reticuloendothelial system (RES) organs, mainly the liver and spleen [33]. In this review, we discuss nanoparticle interactions with the RES and their ultimate distribution to tumors through passive and active targeting, as well as summarize the progress made to bypass RES uptake in nanoparticle chemotherapies. This review strives to analyze (i) the first-pass effect on the delivery of various nanoparticles to solid tumors and (ii) advantages and limitations of evading the RES in nanomedicine delivery. We hope this review points to a direction to overcome current bottlenecks in nanoparticle chemotherapy development.

## 2. A Nanoparticle’s Journey to the Tumor

### 2.1. Barriers and Challenges of Tumor Nanoparticle Uptake

Nanoparticle delivery of poorly soluble drugs has significantly enhanced the therapeutic efficacy of dozens of drugs [34]. Despite this, only a small fraction of passively targeted nanoparticles has been found to accumulate within a solid tumor. Wilhem et al. performed a literature review and calculated that approximately 0.7 percent of injected nanoparticles accumulate within solid tumors in mice [25]. A major caveat of this study is that this is a median value which does not cover the full spectrum of nanoparticle’s potential. Nevertheless, once a nanoparticle is injected into systemic circulation, it must evade several barriers to eventually accumulate in tumor tissue; these barriers include the RES (also known as the mononuclear phagocytic system), which has been implicated in phagocytosis of nanoparticles in liver and splenic tissue [25], as well as rapid renal clearance of small (5–8 nm) particles [26].

Hui et al. elegantly summarized the five major obstacles of nanoparticle delivery to tumors as: opsonization of serum proteins to nanoparticle surfaces, destruction by the immune system, extravasation of nanoparticles into tumors, and finally, infiltration into tumor tissue and the subsequent endocytosis of nanoparticles into individual tumor cells [34]. We have summarized the pathway of nanoparticle delivery and eventual uptake into tumor cells in Figure 1, categorizing the obstacles faced by nanoparticles as either in the vasculature/first-pass organs, extravasation from the vasculature to the tumor tissue, or the internalization of nanoparticles into tumor cells.

The extravasation of nanoparticles from the vasculature into surrounding tumor tissue is a highly studied and extremely complex process. The common belief among researchers is that nanoparticle migration from the vasculature to solid tumor tissue occurs passively through “leaky” gap junctions between endothelial cells lining the vasculature (Figure 2A). Sindhwani et al., however, attempted to correlate frequency/size of gaps between cells with higher nanoparticle uptake and could not find the expected correlation to support this hypothesis [35]. Instead, they determined that the primary contributor to nanoparticle extravasation is an active remodeling process of endothelial cells known as transcytosis (Figure 2B). In comparing wild-type tumor-bearing mice with a model they created that lacks active cellular activity, they determined that active transcytosis is required to facilitate up to 97 percent of nanoparticle uptake into tumor tissue [35].

Cellular internalization of nanoparticles is achieved either passively or actively, although the lines between these two methods of internalization have been somewhat blurred as researchers continue to develop new methods of characterizing and enhancing nanoparticle uptake by tumor cells. Santià et al. reviewed various methods of nanoparticle surface functionalization and found that methods such as PEGylation enhance the passive internalization of nanoparticles [36]. Analytical methods such as transmission electron microscopy (TEM) have been used to visualize accumulation of passively internalized nanoparticles in endosomes, and continued functionalization of nanoparticle coronas with ligands such as cell penetrating peptides (CPPs) have further enhanced nanoparticle uptake into tumor cells. Recent research to functionalize nanoparticles with biomolecules facilitating active uptake include conjugation of antibodies targeting receptors such as HER2 and EGFR, hyaluronic acid-decorated nanoparticles, and aptamers attached to nanoparticles to induce cellular internalization [36].

### 2.2. The Never-Ending Controversy of the EPR Effect

The enhanced permeability and retention (EPR) effect is historically one of the most frequently cited advantages of nanoparticle cancer therapeutics [37]. As tumorigenesis occurs, the secretion of pro-angiogenic factors such as vascular epidermal growth factor (VEGF) induces the formation of a disordered, complex vascular network feeding tumor cells as they proliferate. High expression of VEGF in tumor tissue has been implicated in the formation of abnormal vascular wall morphology, where the endothelium lining the blood vessels contains large gaps—or “leaky” junctions—allowing for enhanced permeability of small particles and cells across the vascular wall. It is this enhanced permeability that is a double-edged sword for cancer: on the one hand, it is believed to enhance tumor metastasis by allowing the escape of tumor cells into the infiltrating vessels, which circulate them across the body to distant metastatic sites [38]; on the other hand, the leaky vasculature unique to the tumor microenvironment promotes the accumulation of systemically circulating nanoparticles at the tumor site, a phenomenon that should enhance the therapeutic efficacy of nanoparticle drugs against solid tumors.

Despite the premise of the EPR effect to preferentially enhance the uptake of nanoparticles in tumors, the challenges we have outlined above clearly indicate there remains significant room for improving nanoparticle delivery. Uptake into tumor tissue via the EPR effect alone has been found to result in a mere 2-fold increase in accumulation of systemically injected drugs [39]. While drugs piggybacking on the EPR effect have excelled in preclinical models, they have yet to find clinical success in humans [40]. For many anti-cancer drugs, such as chemotherapeutics that are inherently toxic, systemic injection of even nanoparticle formulations must be done at very high doses to achieve significant drug accumulation via the EPR effect; this occurs, however, at the cost of off-target toxicity that limits the administered dose and therefore, the therapeutic efficacy of these drugs [39]. In addition, the tumor microenvironment is an inherently complex space with vastly different physiologies within its many compartments, leading to a heterogeneous degree of EPR-based accumulation of drugs in tumors. Danhier summarizes these inherent challenges of the tumor microenvironment for nanomedicine delivery, as well as a summary of the tumor microenvironment’s role on drug uptake [40].

Because of these limitations, we believe there remains a significant need for continued investigation of the mechanisms by which nanoparticles are extravasated into tumor tissue, as recent research indicates this process is more complex than previously understood [41]. New mechanisms have already been recently characterized to enhance our understanding of the EPR effect, such as transcytosis [42]. Furthermore, it is prudent that new strategies continue to be developed that enhance delivery of nanomaterials to tumors with better specificity than that achieved through the EPR effect alone. As we highlight, new engineering strategies to target nanoparticles to tumors, enhance their cellular uptake, and reduce their uptake by the RES are constantly yielding new generations of nanomedicines that can actively enhance tumor delivery of anti-cancer drugs.

## 3. Role of First-Pass Organs on Nanoparticle Delivery

### 3.1. Fates of Different Types of Nanoparticles

A variety of classes of nanoparticles have been developed for drug delivery applications. These include metallic nanoparticles (such as gold, silver, cobalt, and nickel), metal oxide nanoparticles, polymer nanoparticles, lipid nanoparticles, and micelles. Furthermore, many nanoparticle formulations have been further iterated on through functionalization—such as PEGylation—to enhance circulation and delivery. This section provides a broad overview of nanoparticle types as well as a summary of the role of first-pass organs on biodistribution for each nanoparticle type.

#### 3.1.1. Metallic Nanoparticles

Gold nanoparticles (AuNPs) are among the most frequently studied nanoparticles due to their strong biocompatibility [12]. Traditional methods for synthesizing AuNPs, however, frequently result in chemical contaminants or the presence of stabilizing agents used in their manufacture, reducing biocompatibility and increasing toxicity [12]. To circumvent this, Bailly et al. recently developed laser-synthesized AuNPs that exhibit strong biocompatibility and a favorable safety profile. An analysis of biodistribution indicated approximately 0.23 percent of the injected dose accumulated in the kidneys one week after injection, which decreased continually with time. Most of the injected dose (50.4 percent after two weeks) accumulated in the liver and did not decrease, indicating a poor clearance of the AuNPs from these tissues [12]. In a study investigating the role of AuNP size on biodistribution, Takeuchi et al. found that particle sizes of ~100 nm remained in circulation for over 12 h following *i.v.* injection, whereas smaller particles (20–50 nm in diameter) were cleared from systemic circulation within an hour [43]. As was expected from similar studies, all AuNP sizes accumulated preferentially in the liver and spleen, although there was an observed accumulation of AuNPs in both lung and brain tissue, indicating the ability of the AuNPs to cross the blood-brain barrier (BBB). It is hypothesized that this phenomenon occurred due to residual polysorbate-80 stabilizer used in the AuNPs’ manufacture [43].

Silver nanoparticles (AgNPs) have also been employed in numerous cancer drug delivery applications. Silver exhibits strong biocompatibility like gold, but also unique properties such as surface plasmon resonance [44] and, in the field of cancer drugs, induction of dsDNA breaks in cancer cells leading to apoptosis [45]. In tumors, AgNPs have been observed to inhibit the growth of multi-drug resistant MCF-7 human breast cancer cells via inhibition of P-glycoprotein (Pgp) efflux [46]. Gopistty et al. elucidated this mechanism by observing that larger (75 nm) AgNPs were able to inhibit Pgp efflux while small (5 nm) NPs did not, indicating a size-dependence of this therapeutic mechanism [46]. Unlike AuNPs, which inhibit tumor growth primarily through anti-angiogenic effects and promote arrest of cell growth [47], AgNPs have been shown to more directly inhibit tumor growth through inhibition of mitochondrial activity, induction of reactive oxidative species (ROS) production, and the activation of macrophages against tumor cells [47]. Overall, AgNPs are a versatile nanomaterial with the ability to directly act against tumor cells, though additional research is needed to enhance cancer-targeting specificity while minimizing off-target toxicity.

Metal oxide nanoparticles have also been investigated for delivery of cancer drugs. One sub-category of metal oxide nanoparticles includes iron oxide nanoparticles (IONPs) [48]. Alphandéry summarized tumor uptake of IONPs as being approximately 0.0005 to 3 percent of the injected dose for passively targeted nanoparticles, while functionalizing nanoparticles with ligands (molecular/active targeting) increases tumor uptake of nanoparticles to up to 7 percent of the injected dose [48]. An interesting avenue of research pertaining to IONPs is the ability to direct their migration in vivo through the application of an external magnetic field; in their investigation of this technique, however, Alphandéry found that magnetic targeting of IONPs only maximized tumor accumulation to only 2.6 percent of the injected dose, indicating that there is significant room for improving this technique for targeting IONPs to solid tumors.

Other metal oxide nanoparticles include zinc oxide nanoparticles (ZnONPs), which are an interesting nanomedicine platform due to the zinc’s innate ability to act on several molecular pathways such as inducing oxidative stress, increasing cytokine and chemokine secretion, and even selectively inducing apoptosis in various cancer cell lines [49]. In recent years, Bai et al. characterized the ability of ZnONPs to induce apoptosis in SKOV3 human ovarian cancer cells due to oxidative-stress and subsequent DNA damage caused by the nanoparticles [49]. Ancona et al. created an innovative platform that decorated ZnONPs with a DOPC (1,2-dioleoyl-sn-glycero-3-phosphocholine) lipid bilayer, helping to prevent opsonization of serum proteins onto the NPs, increasing their endocytosis into HeLa cells, and also serving as a photodynamic therapy generating ROS when stimulated with UV light [50]. While ZnONPs show an interesting ability to directly arrest cancer cell development and induce cell death, the ROS mechanism by which this occurs means the delivery and action of these nanoparticles must be strictly confined to target cells to minimize off-target toxicity.

Similar to ZnONPs, Titanium dioxide nanoparticles (TiO_2_NPs) are potent photosensitizers and have shown great promise in photodynamic therapies [51]. Çeşmeli and Biray Avci have summarized the various applications of TiO_2_NPs in cancer therapies, particularly emphasizing the ability of these nanoparticles to induce DNA damage through ROS (and subsequent apoptosis of cancer cells), very similarly to ZnONPs. A study by Kongseng et al. also demonstrated the ability of TiO_2_NPs to stimulate secretion of inflammatory cytokines [52]. TiO_2_NPs, like other nanoparticles we have summarized, still exhibit challenges in their delivery. Mainly, this continues to be due to accumulation of nanoparticles in the liver and RES organs, with a majority of nanoparticles being cleared through the renal system [53].

We have summarized various advantages and disadvantages of different metallic nanoparticle systems in Table 2.

#### 3.1.2. Solid Lipid Nanoparticles

Solid lipid nanoparticles (SLNPs) have been frequently investigated for drug delivery in cancer. Muntoni et al. developed SLNPs using fatty acid coacervation that encapsulated methotrexate for delivery to the brain as a glioblastoma treatment [14]. The SLNPs were decorated with free thiols for conjugation of transferrin or insulin, which enhanced permeability and migration across the BBB in a mouse model. The functionalized SLNPs were observed to accumulate significantly (~2–4% of the injected dose) in spleen and liver tissue but did demonstrate enhanced permeability across the BBB. Chirio et al. developed a novel process for the formulation of SLNPs using an oil-in-water microemulsion that allowed for the loading of ~200 nm diameter SLNPs with curcumin, a lipophilic small molecule [15]. In a biodistribution study, the curcumin-loaded SLNPs were found to significantly accumulate in spleen, liver, and lung tissue after one hour, although the concentrations of the nanoparticles in these tissues had been essentially reduced back to their initial concentrations after two hours following *i.v.* administration. These results demonstrate the potential utility of SLNPs for the entrapment and delivery of lipophilic agents and to facilitate their circulation, although continued work is needed to enhance their accumulation and uptake at targeted sites.

#### 3.1.3. Genetically Encoded Micellar Nanoparticles

The engineering of nanomaterials consisting of distinct hydrophilic and hydrophobic regions—or “blocks”—resulted in new self-assembling materials that form micelles when dissolved into solution. We have previously reported on a wide variety of micelles generated using genetically encoded biopolymers, where polypeptides such as elastin-like polypeptides (ELPs) can be designed with hydrophobic domains to form micelles encapsulating lipophilic or hydrophobic drugs [5]. Within this vast field of genetically encoded polymer nanoparticles, some prominent examples include the work of MacKay et al. and Bhattacharyya et al., who developed chimeric polypeptide (CP) nanoparticles containing drug attachment domains for chemotherapeutics such as doxorubicin (DOX) and paclitaxel (PTX), respectively [54,55]. CP-chemotherapeutics demonstrated significant anti-tumor efficacy in vivo, enhancing plasma circulation of the drugs and minimizing off-target toxicity.

Yousefpour et al. enhanced the therapeutic efficacy of CP-chemotherapeutic nanoparticles by developing a CP fused to an albumin-binding domain (ABD) and conjugated DOX to this fusion protein, creating nanoparticles that bound endogenous albumin upon *i.v.* injection and circulated for a significant time in vivo [56]. Furthermore, the ABD-CP-DOX nanoparticles exhibited less accumulation in liver and splenic tissue than naked CP-DOX nanoparticles, indicating the utility of functionalizing these nanoparticles with endogenous serum proteins to reduce accumulation in RES organs.

#### 3.1.4. Other Polymeric Nanoparticles

Other polymer nanoparticles have demonstrated unique and promising attributes advantageous for cancer drug delivery. Such nanoparticles include dendrimers, which are unique for their polymer backbone but highly branched structure [20]. Carvalho et al. summarized the many applications for dendrimer nanoparticles in cancer drug delivery [20]. Salimi et al. synthesized dendrimer-IONPs that demonstrated a steady accumulation and retention in kidney tissue, while an initial accumulation of IONPs in liver tissue decreased over 12–24 h following *i.p.* injection [21]. This decrease was hypothesized to occur due to IONPs accumulating in the spleen and lymph nodes, but the concentration of nanoparticles in these compartments was not measured in the study.

Other polymer nanoparticles have been investigated using biodegradable polymers such as poly(lactide-*co*-glycolide), or PLGA. Rafiei and Haddadi synthesized and characterized the biodistribution and pharmacokinetics of PLGA nanoparticles loaded with docetaxel and modified with polyethylene glycol (PEG) [22]. They found that PEG-PLGA nanoparticles exhibited significantly less accumulation in liver, kidney, heart, and lung tissue compared to naked PLGA nanoparticles in mice treated *i.v.* with the docetaxel-loaded nanoparticles. Furthermore, PEG-PLGA nanoparticles exhibited a longer, sustained cumulative release of docetaxel over five days following treatment compared to naked PLGA nanoparticles. This example demonstrates the utility of tools such as PEGylation to improve pharmacokinetics and biodistribution of drugs in polymer nanoparticle systems.

We have summarized the advantages and disadvantages of the various polymeric nanoparticle systems in Table 3.

#### 3.1.5. Next-Generation Nanoparticle Systems

Several advanced technologies are being developed to help surmount issues that have prevented current nanotechnologies from exhibiting significant clinical benefits. Such strategies involve combining molecular targets for cancer cell signaling pathways with nanoparticles decorated with targeting moieties to enhance delivery to tumors. One example is the work of Singh et al., who developed a prostate cancer therapeutic using planetary ball-milled nanoparticles (PBM-NPs) coated with a prostate-specific membrane antigen-binding RNA aptamer [57]. They found that these aptamer-decorated PBM-NPs could encapsulate and efficiently deliver thymoquinone (an inhibitor of Hedgehog protein signaling) to C4-2B and LNCaP prostate cancer cells and inhibit cancer cell proliferation in vitro. Mukherjee recently developed nanoparticles consisting of a silver Prussian blue analogue (Ag_3_[Fe(CN)_6_]), with the nanoparticles they synthesized exhibiting both antimicrobial and anti-cancer properties (testing against various Gram-negative and -positive bacteria, as well as tumor models such as B16F10) [58]. More recently, Mukherjee et al. developed novel PEGylated platinum nanoparticles allowing for the loading of DOX as a treatment for melanoma in mice [59]. They found that the PEGylated platinum nanoparticles could efficiently deliver DOX *i.p.* to B16F10 and A549 tumor-bearing mice, with the nanoparticle drug exhibiting greater therapeutic efficacy compared to a DOX control.

Other approaches include novel nanomaterials comprised of hybrid organic/inorganic materials. Pan et al. developed a novel mesoporous silica nanoparticle (MSN) bound to poly(oligo(ethylene glycol) monomethyl ether methacrylate) (POEGMA) to enhance circulation and impart stealth behavior [60]. They further decorated these MSNs with an integrin binding domain (RGD targeting peptide) and found that the RGD-POEGMA-MSNs were efficiently internalized by HCT116 human colon cancer cells and, when these MSNs were loaded with 5-fluorouracil (a common anti-cancer agent), a significant portion of the MSNs accumulated in the tumors of HCT116 tumor-bearing mice through 48 h after *i.v.* injection [60]. The MSNs exhibited promising anti-cancer activity, demonstrating the potential clinical utility of this approach. Despite this, however, there was still significant accumulation of the MSNs in liver tissue and some observed liver toxicity in mice, indicating there remains significant work to be done to tune the delivery of these MSNs for patient use.

Recent advances in cancer nanomedicines have also yielded innovative combinations of traditional chemotherapeutics with targeting/therapeutic antibodies. Abedin et al. recently developed a novel approach in which nanorods are conjugated to hydrophobic PTX (PTXNRs) and formed into nanoparticles decorated with Trastuzumab (TTZ), an FDA-approved HER2 targeting therapeutic antibody [61]. These PTXNR-TTZs were found to exhibit synergism, inhibiting a greater percentage of various breast cancer cell lines in vitro (including BT-474 and SK-BR-3) than a combination of PTX and TTZ administered separately. Further investigation of this novel drug formulation is needed to determine the effectiveness of this approach at targeting the nanoparticles to tumor tissue in vivo, but these approaches represent promising new methods for enhancing the delivery of nanomedicines to tumors.

### 3.2. Role of Size and Surface Chemistry

Upon administration, nanoparticles interact with elements of the physiological environment such as blood, interstitial fluid, extra-cellular matrix, and cellular cytoplasm—all of which contain a complex mixture of proteins that adsorb onto the surface of nanoparticles, forming a protein corona [62]. This protein corona changes the inherent synthetic identity of nanoparticles and is primarily responsible for their rapid clearance from the physiological environment through first-pass metabolism [30]. Many of these proteins mark the nanoparticle for efficient clearance by the RES (opsonization). Generally, larger nanoparticles accumulate in the liver and spleen more rapidly. In a seminal work, Walkey and co-workers systematically analyzed the effect of size and surface chemistry of a nanoparticle on serum protein adsorption and effective phagocyte evasion [13]. Using label-free liquid chromatography tandem mass spectrometry, they identified 70 different serum proteins that are heterogeneously adsorbed to the surface of model gold nanoparticles. The relative density of each of these adsorbed proteins depends on nanoparticle size and PEG grafting density [13]. At a fixed PEG grafting density, decreasing nanoparticle size increases total protein adsorption. In another study, a significantly lower percentage of an injected dose of 10 nm Al_2_O_3_ nanoparticles accumulated in the liver compared to all larger nanoparticles ranging from 40 to 10,000 nm. There is still debate as to whether the rapid accumulation is due to simple filtration or increased binding opportunities between the RES cells and nanoparticles. Yousefpour et al. recently demonstrated the effects of albumin binding on tumor accumulation and liver uptake of polypeptide nanoparticles. Albumin binding decreased liver uptake of nanoparticle-bound DOX by 1.5-fold and improved tumor accumulation by 2.5-fold [56].

### 3.3. Role of Dosing

The effect of nanoparticle dose has not been carefully and systematically investigated until recently. Analyzing the effects of dosing across different classes of nanoparticles (such as AuNPs) in literature is difficult, as researchers frequently report only the therapeutic’s dose, as opposed to also reporting the dose of the nanoparticle itself. There is no standardized metric for nanoparticle dosing. In the example of a CP nanoparticle, dose affects the tumor and liver accumulation of the DOX payload. As DOX is covalently conjugated to the CP, DOX concentration is directly proportional to nanoparticle concentration. At a 20 mg/kg dose of DOX, the liver accumulation of DOX decreased compared to a 10 mg/kg dose [56]. However, the tumor accumulation increased proportionally to DOX dose. The surface coating of the CP-DOX nanoparticle with an albumin-binding peptide also did not change this observation (Figure 3). Ouyang et al. identified a dose threshold—one trillion nanoparticles in mice—for improving nanoparticle delivery to tumors [63]. Importantly, this dose threshold saturates the rate of nanoparticle uptake by Kupffer cells. This study has the potential to establish a standardized metric that could be used across various nanoparticles and animal species.

## 4. Understanding Opsonization of Proteins onto Nanoparticles

The interaction of nanoparticles with physiological systems—the so-called “nano-bio interface”—is a complex process. It is therefore important to understand this mechanism in some detail before we delve into strategies to bypass first-pass metabolism. As we discussed, a nanoparticle’s inherent composition, shape, size, and surface chemistry plays a crucial part in deciding its fate in systemic circulation. In this section, we discuss the role of two external parameters that play equally vital roles to affect circulation stability and the final disposition of a nanoparticle to the tumor: (i) formation of a protein corona and (ii) the mechanism of interactions between the nanoparticle and cells.

When a nanoparticle enters the blood, it is coated with various serum proteins, forming a protein corona that changes the nanoparticle’s synthetic identity, exposes new epitopes, and alters its function. The concept of a protein corona is thus an important parameter in shaping the final biological identity of the nanoparticle—hydrodynamic size, surface chemistry, net charge, and aggregation behavior [30,32,64]. Although this has been known for a long time, it is only recently that researchers were able to decipher the complex mixture of adsorbed proteins on nanoparticles. This was mainly led by the advancement of instrumentation and analytical methods, coupled with the motivation to shake the existing stagnancy of clinical translation for nanomedicine delivery systems [65,66,67]. Albumin, apolipoprotein, immunoglobulins, transferrin, fibrinogen, complement C3, haptoglobin, and α-2-macroglobulin are some of the most abundant proteins that comprise the protein corona [32,68]. Nanoparticles interact with these proteins mainly through long range electrostatic Van der Waal’s forces, as well as short-range hydrophobic interactions [33]. There are two types of proteins in the protein corona [65,66]: opsonin and dysopsonin. Adsorption of opsonin at the nanoparticle surface results in its recognition by mononuclear macrophages, which rapidly clear it from circulation through first-pass metabolism. Adsorption of dysopsonin, on the other hand, has the opposite effect—it prolongs the circulation of a nanoparticle. The formation of a protein corona is a dynamic process and is affected by the affinity of the proteins to nanoparticles, as well as by the protein concentration in a biological medium. The hard corona is the inner layer, irreversibly bound to nanoparticle surface, and exchanges with the physiological medium within a matter of hours. The soft corona is the outer layer, reversibly bound to nanoparticle, and exchanges with the physiological medium on a timescale of seconds to minutes [67]. Since the hard corona remains bound to the surface until the degradation of the nanoparticle, it plays a more profound role than the soft corona in governing the downstream processing of the *i.v.* administered nanoparticle, such as endocytosis and translocation to different organs. The composition of the protein corona undergoes constant changes in circulation: albumin and fibrinogen, proteins that are abundant in serum, dominate the composition of the protein corona of an *i.v.* administered nanoparticle for a short period of time. In the long run, relatively scarce proteins with higher affinities and slower kinetics—such as apolipoprotein—may replace them [31]. The total amount of adsorbed protein, however, remains relatively constant [32].

Formation of a protein corona not only changes the pharmacokinetics, but also the pharmacodynamics of a nanoparticle by affecting its interaction with the various cells and subcellular organelles. As is the case for all foreign substances, nanoparticles are cleared from the bloodstream by cells of the RES. This part of the immune system consists of phagocytes such as monocytes and macrophages, which are mainly located in the liver, spleen, lungs, and lymph nodes. Phagocytosis of nanoparticles is promoted by opsonizing proteins, such as immunoglobulins and complement proteins, which tag the nanoparticles as a foreign substance and facilitate their recognition by the RES [69,70]. In a seminal work, Deng et al. demonstrated that negatively charged poly(acrylic acid)-conjugated gold nanoparticles mostly bind to fibrinogen, exposing the γ377–395 chain. This conformational change promotes interaction of the protein with the integrin receptor Mac-1. Activation of this receptor turns on the NF-κB signaling pathway, resulting in the release of inflammatory cytokines and thereby facilitating the recognition of the nanoparticle by RES cells [71]. Coating the nanoparticle surface with TNF-α alters the interaction of the nanoparticle with fibrinogen and decreases the rate of blood clearance [72]. Resident macrophages were primarily responsible for the bulk of nanoparticle uptake in the liver, while spleen uptake was highly surface property dependent. In another work, Vogt et al. compared protein corona formation and macrophage uptake of silica-coated and dextran-coated superparamagnetic iron oxide nanoparticles (SPIONPs) [73]. They made a comprehensive list of proteins comprising the protein corona on those nanoparticles by using gene ontology (GO) enrichment analysis and Kyoto Encyclopedia of Genes and Genomes (KEGG) pathway analysis. The corona was shown to promote macrophage uptake of silica-coated, but not dextran-coated, nanoparticles. Mohammapdour et al. comprehensively summarized the cellular mechanisms of interaction between inorganic nanoparticles and different immune cells, including macrophages [21]. It is important to note that adsorption of proteins can trigger conformational changes that result in a loss of functionality, exposure of cryptic epitopes, and adverse immune responses [65].

## 5. Strategies to Circumvent First-Pass Metabolism

Surface modification of nanoparticles with PEG, also known as “PEGylation,” has long been a standard approach in nanomedicine to reduce phagocytosis and improve tumor accumulation of nanoparticles [74]. PEGylation increases the hydrodynamic radius of a nanoparticle beyond the renal filtration cut-off and shields immunogenic epitopes of the nanoparticle, preventing its clearance by RES organs. However, accumulating evidence suggests it is necessary to find an alternative to PEGylation, as it has significant shortcomings. First, PEGylation lowers uptake by target cells [75]. Second, PEG induces a significant anti-PEG antibody response upon treatment with PEGylated therapeutics [76]. Because of this phenomenon, PEGylated nanoparticles have a reduced circulation time before they are cleared [76,77]. Moreover, ~67% of the US population (who have never been administered PEGylated drugs) have been found to have pre-existing anti-PEG antibodies [78], likely due to the ubiquitous use of PEG in excipients, laxatives, and other various consumer products. The high titer of induced and pre-existing anti-PEG antibodies can compromise the clinical efficacy of PEGylated nanoparticles and result in life-threatening anaphylactic reactions [77], as seen most recently in the COVID-19 lipid nanoparticle (LNP)-mRNA vaccines. In response to these challenges, the FDA now requires special monitoring of clinical trials that administer PEGylated drugs [79]. To address the shortcomings of PEG, Ozer et al. developed a PEG-like brush polymer in which the long, immunogenic PEG sequence is broken into shorter oligoethylene glycol oligomers and stacked as side-chains on a poly(methyl methacrylate) backbone [80]. The brush polymer did not induce anti-PEG antibody binding in vivo and retained the favorable traits of a traditional PEG system. Banskota et al. developed a zwitterionic polypeptide (ZIPP) as an alternative to PEG, which contained a pentameric repeat unit of an elastin-like polypeptide precursor. The pentameric unit contains 1:1 ratios of amino acids with positively and negatively charged residues. This ZIPP formed nanoparticles triggered by conjugation of multiple copies of hydrophobic PTX molecules (ZIPP-PTX), and this drug achieved a two-fold increase in tumor accumulation. They did not, however, report on the liver uptake of ZIPP-PTX in this study [16].

RES blockade is another strategy that saturates, blocks, or depletes macrophages to boost efficacy of nanoparticle therapeutics by salvaging them from opsonization. In 1983, a seminal work by Proffitt inspired many researchers to use conventional blank liposomes to saturate the RES in various preclinical settings [81]. Several reports support the notion that high doses of liposomes can overwhelm the RES and increase tumor accumulation of nanoparticles [82]. For example, Liu et al. temporarily blocked the RES using a commercial liposome that increased tumor accumulation of small sized, PEGylated nanoparticles [83]. This approach is also clinically attractive as it involves administering nontoxic phospholipids. This effect is only temporary, however, and the dose amount and interval must be carefully optimized, necessitating repeated injections for every treatment schedule.

RES depletion with unique chemical agents has also gained traction in recent years. Gadolinium chloride (GdCl_3_) suppresses RES activity and selectively eliminates the large Kupffer cells in the liver. Diagaradjane et al. reduced nonspecific sequestration of quantum dots (QDs) by RES macrophages by pretreating mice with GdCl_3_, increasing circulation time and amplifying the tumor-specific signal of conjugated QDs [84]. The anti-malarial drug chloroquine is also used to reduce clearance of nanoparticles by macrophages [85]. This has led to improved tumor accumulation of various nanoparticle therapeutics. Opperman et al. recently demonstrated that clodronate-liposome administration resulted in depletion of CD169^+^ bone marrow–resident macrophages [86]. Methyl palmitate [87], dextran sulfate [88], and carrageenan [89] can also serve as chemical tools to deplete phagocytic liver cells. However, there are two major limitations of this approach. First, their administration is limited by systemic toxicity. Second, even though depletion of phagocytic cells leads to 18–120 times greater delivery efficiency of the nanoparticles to a solid tumor, only 2 percent of the injected nanomaterials accumulated in the tumor tissue [90]. Tang et al. used liposomes decorated with CD47 (a membrane glycoprotein expressed on mammalian cells that gives phagocytes a “don’t-eat-me” signal) to block RES uptake and subsequently improve delivery of PLGA nanoparticles [91].

## 6. Outlook and Conclusions

Nanoparticles have no doubt revolutionized the delivery of drugs for cancer. Our repertoire of available treatments has greatly expanded in efficacy thanks to innovative delivery systems that encapsulate and release therapeutics using nanoparticles. Despite these advances, it is clear that significant improvements are needed in the targeted, sustained delivery of nanomedicines to tumors. New materials engineering strategies that we have highlighted, including functionalization with proteins, PEGylation, stimuli-responsiveness, and bio-engineered designs, will continue to reshape how nanomedicines treat cancer. Specifically, we see several necessary areas of research to improve the effectiveness of nanoparticle-based cancer therapeutics. First, the proportion of a nanomedicine’s injected dose that ultimately reaches the tumor must be drastically improved. As we have highlighted, reliance on the EPR effect alone, or even as a peripheral factor, is not enough to achieve significant accumulation of nanomedicines at a tumor site. Between the preferential uptake of nanoparticles by the RES organs, renal clearance of particles, and poor extravasation of particles into tumor tissue from the vasculature, current solutions result in only a very small percentage of nanoparticles reaching the tumor site. New, “smart” materials that have been engineered with stealth behavior to improve circulation, reduce RES uptake, and enhance transcytosis into tumor tissue will no doubt represent the future of clinically successful nanomedicines. Second, the contribution of liver to deplete the nanoparticle concentration in circulation may have been overestimated. We need to look beyond the RES and systematically describe the interactions between nanoparticles and other physiological barriers to facilitate delivery of nanoparticles to tumors. Careful mapping of such interactions will enable better design of nanomaterials to overcome specific physiological barriers. Third, as we have described, dose strategies will need to be carefully considered by researchers to maximize the therapeutic window of nanoparticle drugs. Dose-limiting toxicity of chemotherapeutics remains a challenge even in today’s most advanced nanomedicines. Strategies to overcome this may be centered around targeting modalities that can be employed to deliver and retain nanoparticle drugs to a tumor site to maximize local tumor toxicity more precisely. One potential application is the use of external magnetic fields to direct IONPs to local tumor sites, a concept which we have shown needs significant research to achieve clinical utility. Despite the challenges the field faces, we believe that the future of nanomedicines for cancer therapeutics is as bright as ever. As new technologies are developed to overcome the challenges we have highlighted in this review, the efficacy of nanoparticle drugs will undoubtedly cross an inflection point leading to tremendous efficacy in the future. Ultimately, we see many future cancer treatments successfully employing nanoparticles to hopefully improve patient outcomes across a broad range of cancers.

## Figures and Tables

**Figure 1 cancers-14-01741-f001:**
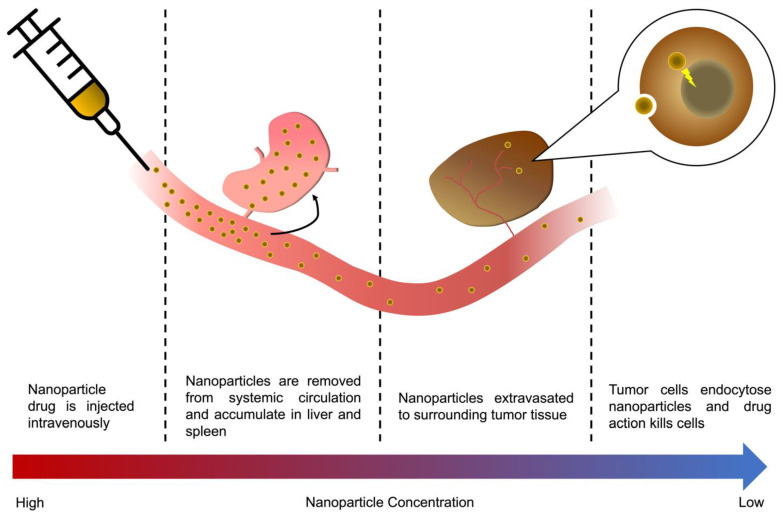
A nanoparticle’s journey to the tumor, from injection to mode of action. The majority of an *i.v.* injected nanoparticle drug will, in a matter of hours, be metabolized by the RES organs, accumulating in the liver and spleen, or be cleared through renal clearance depending on particle size. Those particles that are not cleared through RES uptake or renal clearance will eventually accumulate in the tumor vasculature are extravasated to surrounding tissue, either via leaky vessels in the tumor or active transcytosis across the vascular wall. Eventually, a small fraction of the original dose is endocytosed into tumor cells, killing them.

**Figure 2 cancers-14-01741-f002:**
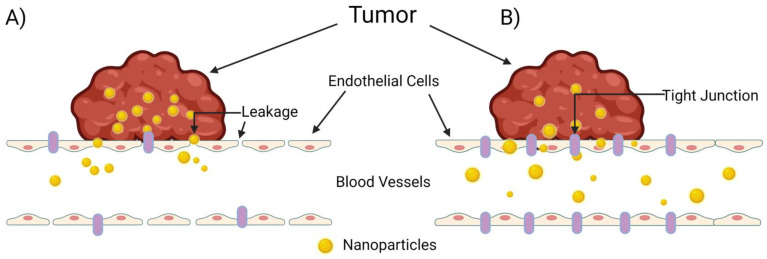
The mechanisms of nanoparticle entry into solid tumors. The escape of a nanoparticle from circulation into a solid tumor plays a pivotal role on a nanoparticle’s overall therapeutic efficacy, but this process is poorly understood. (**A**) The enhanced permeability and retention (EPR) effect, in which the nanoparticle enters the tumor through leaky tumor vasculature, and (**B**) transcytosis of the nanoparticle through intra-endothelial channels or vesicles in the absence of leakage between endothelial cells, as proposed by Sindhwani et al. [35]. Created with Biorender.com (accessed on 27 March 2022).

**Figure 3 cancers-14-01741-f003:**
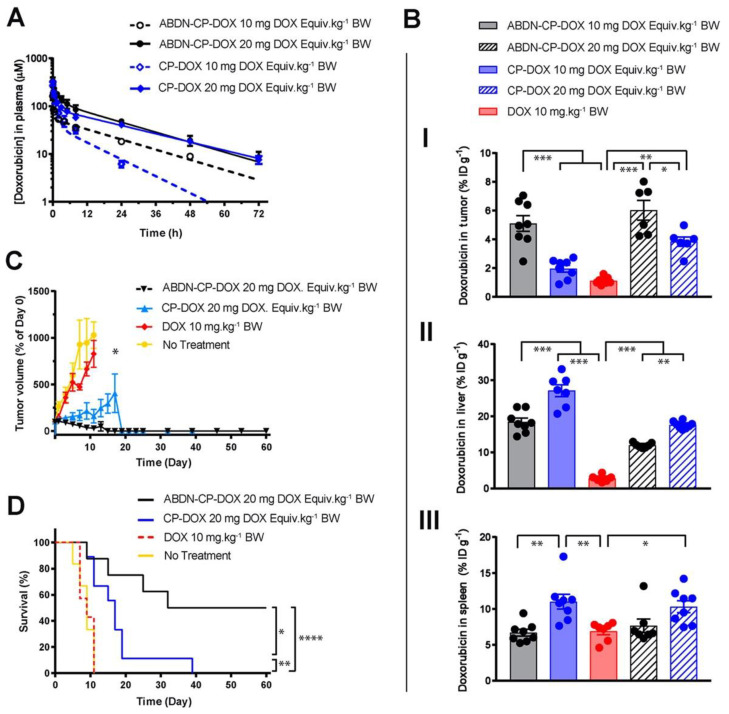
Role of dosing on nanoparticle drug efficacy. An albumin binding nanoparticle of doxorubicin (ABDN-CP-DOX) was compared head-to-head with a non-albumin binding counterpart (CP-DOX). (**A**) pharmacokinetic profile of the nanoparticles at different doses in mice. (**B**) Biodistribution of the doxorubicin conjugated to various nanoparticles at 24 h post-administration in the (I) tumor, (II) liver, and (III) spleen. All nanoparticles show better tumor accumulation and less liver accumulation at the higher dose. (**C**) Tumor regression curve in an *s.c.* mouse C26 colon cancer model, up to day 60. (**D**) cumulative survival of tumor-bearing mice treated with indicated drugs. * for *p* < 0.05, ** for *p* < 0.01, *** for *p* < 0.001, and **** for *p* < 0.0001. Reprinted (adapted) with permission from [56]. Copyright 2018 American Chemical Society.

**Table 1 cancers-14-01741-t001:** Active advanced stage clinical trials of nanoparticle cancer drugs in the United States.

Phase	Clinical Trial Number	Nanoparticle Type	Active Payload	Conditions
Phase 4	NCT04258631	Liposome	Bupivacaine	Malignant female reproductive system neoplasms
Phase 3	NCT04033354	Nab-paclitaxel	Paclitaxel	Squamous non-small cell lung cancer
	NCT00785291	Nab-paclitaxel	Paclitaxel	-
	NCT00108735	Paclitaxel-polyglumex		Fallopian tube and ovarian cancer
	NCT03768414	Nab-paclitaxel	Paclitaxel	Bile duct and gallbladder cancer
	NCT02839707	Pegylated Liposome	Doxorubicin	Fallopian tube and ovarian cancer
	NCT02580058	Pegylated Liposome	Doxorubicin	Ovarian cancer
	NCT03197935	Nab-paclitaxel	Paclitaxel	Triple-negative breast cancer
	NCT03941093	Nab-paclitaxel	Paclitaxel	Non-resectable pancreatic cancer
	NCT03088813	Liposome	Irinotecan	Small cell lung cancer
	NCT02101788	Pegylated Liposome	Doxorubicin	Borderline ovarian serous tumors
	NCT03257033	Nab-paclitaxel	Paclitaxel	Locally advanced pancreatic cancer
	NCT04895358	Nab-paclitaxel	Paclitaxel	Breast neoplasms
		Pegylated Liposome	Doxorubicin	
	NCT01964430	Nab-paclitaxel	Paclitaxel	Pancreatic neoplasms

**Table 2 cancers-14-01741-t002:** Advantages and Disadvantages of Metallic and Metal Oxide Nanomaterials for Cancer Therapies.

Nanoparticle Class	Advantages	Disadvantages
Gold Nanoparticles (AuNPs)	Strong biocompatibility	Chemical contaminants from synthesis can cause toxicity issues
	Established delivery platform for a variety of cancer drugs	Less direct anti-cancer effects than other nanoparticle materials
Silver Nanoparticles (AgNPs)	Good biocompatibility	Size-dependent cytotoxicity requires tuning of particle size
	Direct anti-cancer cell killing capability	Potential off-target effects with little delivery to the tumor
Iron Oxide Nanoparticles (IONPs)	Ability to direct uptake through external magnetic stimulation	Active targeting requires significant research to achieve clinical utility
	Can be functionalized with ligands to enhance active targeting	
Zinc Oxide Nanoparticles (ZnONPs)	Innate action on molecular pathways inducing ROS, cytokine and chemokine secretion, and cancer cell apoptosis	Off-target effects with poor tumor accumulation must still be addressed in vivo
	Cytotoxic effects can be tied to external stimulation, such as UV light	
Titanium Dioxide Nanoparticles (TiO_2_NPs)	Similar direct cytotoxicity mechanisms as ZnONPs, through ROS generation and DNA damage to cancer cells	NPs frequently accumulate in RES organs are cleared through the renal system before significant tumor accumulation

**Table 3 cancers-14-01741-t003:** Advantages and Disadvantages of Polymeric Nanoparticles for Cancer Therapies.

Nanoparticle Class	Advantages	Disadvantages
Solid Lipid Nanoparticles (SLNPs)	Controlled synthesis using oil-in-water microemulsions	Majority of SLNPs accumulate in liver and spleen tissue
	Optimal for loading of lipophilic agents	
Micellar Chimeric Polypeptide Nanoparticles (CP-NPs)	Significant anti-cancer toxicity with minimal off-target toxicity	Synthesis of CP-NPs requires synthesis in *Escherichia coli* and purification of endotoxin prior to administration
	Ease of synthesis with targeting domains and peptides encoded at the gene level into a fusion protein with the CP-NPs	
Dendrimers	Ability to synthesize nanoparticles with targeting ligands for cancer therapies	Accumulation in kidney tissue and likely RES organs reduces anti-cancer efficacy
PEG-PLGA Nanoparticles	Significantly less accumulation in liver, kidney, heart, and lung tissue than other nanoparticle systems	Allergic reactions due to anti-PEG antibodies may limit widespread use
	Simple synthesis and encapsulation of chemotherapeutics	

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
