# Peer review of "A Nanoparticle’s Journey to the Tumor: Strategies to Overcome First-Pass Metabolism and Their Limitations"

_cancers, 2022, doi:10.3390/cancers14071741_

Round 1

Reviewer 1 Report

In this review, the authors describe the challenges encountered by anticancer nanomedicines, specifically those related to first pass metabolism and clearance mechanisms that markedly limit their ability to reach the target tumour site. The scope of the article is timely and highly relevant, providing insights into how tumour deposition of nanoparticles could be improved to expedite their clinical translation. Further revisions as suggested below will render this article suitable for publication in Cancers, appealing to both the readership of the journal and also the wider nanomedical field.

Major edits:

  • The review needs to be more comprehensive. At present, it is quite brief and could benefit from, for example, additional detail around the mechanisms responsible for nanoparticle clearance/metabolism. A new section outlining mechanisms such as the RES/renal system in-depth would be highly useful, perhaps alongside a Figure for improved clarity.
  • In its current format, it is clear that different sections of the article have been written by different authors. In particular, the writing style within the summary/abstract/introduction appears very different from the rest of the article and also a little confusing and disjointed in places. The article would benefit from significant revisions to the summary/abstract/introduction to improve upon grammar, clarity and flow of the text, which will make it appear more coherent and comprehensible to the reader.

Minor edits:

  • References are missing in places. For example, in lines 48-49, the authors state that ‘rapid clearance of circulating NP by first-pass effect has been identified as the major contributor for such dismal performance’. However, no references are provided to support this statement. Likewise, a reference is needed for lines 277-279 etc. In general, more references are needed throughout the article.
  • The article would greatly benefit from the following Figure edits: (1) additional detail within Figure legends 1 and 2, (2) improved aesthetics of Figure 1 and (3) improved quality of Figures 2 and 3, which appear blurred / of low resolution at present.
  • Abbreviations need to be defined at first mention in the text. Consistency needed in places e.g. both RES and MPS used interchangeably.
  • Lines 100-101 ‘The primary contributor to nanoparticle is an active…’ The phrase is missing a word.

Author Response

Reviewer 1

In this review, the authors describe the challenges encountered by anticancer nanomedicines, specifically those related to first pass metabolism and clearance mechanisms that markedly limit their ability to reach the target tumour site. The scope of the article is timely and highly relevant, providing insights into how tumour deposition of nanoparticles could be improved to expedite their clinical translation. Further revisions as suggested below will render this article suitable for publication in Cancers, appealing to both the readership of the journal and also the wider nanomedical field.

Major edits:

  • The review needs to be more comprehensive. At present, it is quite brief and could benefit from, for example, additional detail around the mechanisms responsible for nanoparticle clearance/metabolism. A new section outlining mechanisms such as the RES/renal system in-depth would be highly useful, perhaps alongside a Figure for improved clarity.

We thank the reviewer for this important comment. We have discussed nanoparticle’s interaction with the biological interface and their clearance through RES in section 4.

  • In its current format, it is clear that different sections of the article have been written by different authors. In particular, the writing style within the summary/abstract/introduction appears very different from the rest of the article and also a little confusing and disjointed in places. The article would benefit from significant revisions to the summary/abstract/introduction to improve upon grammar, clarity and flow of the text, which will make it appear more coherent and comprehensible to the reader.

We thank the reviewer for the critical observation. We have tried our best to organize and maintain the flow of logic in the revised manuscript.

Minor edits:

  • References are missing in places. For example, in lines 48-49, the authors state that ‘rapid clearance of circulating NP by first-pass effect has been identified as the major contributor for such dismal performance’. However, no references are provided to support this statement. Likewise, a reference is needed for lines 277-279 etc. In general, more references are needed throughout the article.

We have cited appropriately through out the manuscript in our revised version.

  • The article would greatly benefit from the following Figure edits: (1) additional detail within Figure legends 1 and 2, (2) improved aesthetics of Figure 1 and (3) improved quality of Figures 2 and 3, which appear blurred / of low resolution at present.

We have updated all the figures and legends.

  • Abbreviations need to be defined at first mention in the text. Consistency needed in places e.g. both RES and MPS used interchangeably.

We thank the reviewer for pointing out this. We have incorporated the suggestion in the revision.

  • Lines 100-101 ‘The primary contributor to nanoparticle is an active…’ The phrase is missing a word.

Apologies for the typo. We have now fixed it.

Reviewer 2 Report

Despite improvements in cancer research and chemotherapy development, cancer remains a typical and dangerous disease. In fact, tumor heterogeneity, medication resistance, and systemic toxicity are significant challenges in cancer treatment. Further, off-target toxicity is a significant problem with traditional cancer therapies, limiting their effective dose and preventing patients' malignancies from being adequately treated with chemotherapeutics alone. The safe administration of chemotherapeutics with side effects was made possible by nanomedicine, in which a medication is bundled into a nanoparticle that shields it from peripheral locations until the tumor takes it up. Because of their potential for targeting and multifunctionality, nanoscale delivery systems, or nanotherapies, are becoming more popular as vehicles for antineoplastic medicines. Despite the benefits of nanomedicines for cancer, improving tumor absorption and preventing premature clearance from the body remain substantial problems. In this study, the authors explore the potential ways for using nanotherapies to overcome difficulties in cancer treatment. In this review, they describe the impact of first-pass metabolism on nanoparticles' route to the tumor. The article is well written, and I enjoy reading it. However, I have a few concerns in this article.

There are numerous publications and review articles on these nanoparticles accessible (AuNPs, SLNPs, Micelle-NPs). I advise the authors is to focus on a few advanced technologies employed in nanotherapy and their limitations, such as nucleic acid aptamers, antibody therapy, PBM-NPs therapy, and so on. See the articles published by Singh et al. Cells, 2020 on PBM-NPs; Pan et al. 2017 etc.

I'm not sure why the authors mention gold and iron oxide nanoparticles. If they're talking about metallic NPs, they should also write about Silver NPs.

Despite focusing on in vivo first-pass metabolism, the authors did not investigate the molecular pathways or targets of those NPs. In all NPs studies, the authors should address these. This article is short due to the paucity of molecular targets studied in nano-therapy.

They should also consider the role of immune checkpoint blockade and drug resistance during nanoparticles therapy.

The authors should include tables with the advantages and disadvantages of these nanoparticles.

The clinical trials information’s on these NPs should be added.

Although the authors included a reference 47, the legends in Figure 3 are poorly stated.

Author Response

Despite improvements in cancer research and chemotherapy development, cancer remains a typical and dangerous disease. In fact, tumor heterogeneity, medication resistance, and systemic toxicity are significant challenges in cancer treatment. Further, off-target toxicity is a significant problem with traditional cancer therapies, limiting their effective dose and preventing patients' malignancies from being adequately treated with chemotherapeutics alone. The safe administration of chemotherapeutics with side effects was made possible by nanomedicine, in which a medication is bundled into a nanoparticle that shields it from peripheral locations until the tumor takes it up. Because of their potential for targeting and multifunctionality, nanoscale delivery systems, or nanotherapies, are becoming more popular as vehicles for antineoplastic medicines. Despite the benefits of nanomedicines for cancer, improving tumor absorption and preventing premature clearance from the body remain substantial problems. In this study, the authors explore the potential ways for using nanotherapies to overcome difficulties in cancer treatment. In this review, they describe the impact of first-pass metabolism on nanoparticles' route to the tumor. The article is well written, and I enjoy reading it. However, I have a few concerns in this article.

There are numerous publications and review articles on these nanoparticles accessible (AuNPs, SLNPs, Micelle-NPs). I advise the authors is to focus on a few advanced technologies employed in nanotherapy and their limitations, such as nucleic acid aptamers, antibody therapy, PBM-NPs therapy, and so on. See the articles published by Singh et al. Cells, 2020 on PBM-NPs; Pan et al. 2017 etc.

We thank the reviewer for this valuable comment. We have added a section (3.1.5) on next-generation nanoparticles and included the suggested references.

I'm not sure why the authors mention gold and iron oxide nanoparticles. If they're talking about metallic NPs, they should also write about Silver NPs.

We thank the reviewer for pointing this out. We have now discussed other metallic nanoparticles such as silver, zinc oxide, titanium dioxide in section 3.1.1.

Despite focusing on in vivo first-pass metabolism, the authors did not investigate the molecular pathways or targets of those NPs. In all NPs studies, the authors should address these. This article is short due to the paucity of molecular targets studied in nano-therapy.

We thank the reviewer for bringing this up. We have now added molecular targeting of the nanoparticles in section 3.1.5. We have also discussed in detail interaction of such nanoparticles with the biological interface in section 4.

They should also consider the role of immune checkpoint blockade and drug resistance during nanoparticles therapy.

We thank the reviewer for this interesting suggestion. We also agree this is a very promising development in the field of nanomedicine. However, we believe this is out of the scope of the present manuscript.

The authors should include tables with the advantages and disadvantages of these nanoparticles.

Thanks for this important suggestion. We have included advantages and disadvantages of different nanoparticles in Table 2 and Table 3

The clinical trials information’s on these NPs should be added.

Again, we thank the reviewer for this valuable suggestion. We have included this info in table 1.

Although the authors included a reference 47, the legends in Figure 3 are poorly stated.

We have now updated all the figure legends to include more details.

Reviewer 3 Report

The authors have attempted to do a quick review on the current challenges faced by nanoparticles such as first-pass metabolism and how to solve them. In general, the manuscript requires extensive grammar check and English editing. Additionally, please see some comments below:

1) line 92: Editing required...“Nanoparticles that make it to…”

2) Figure 2: the image resolution is very poor such that the image and text in the image isn’t clearly visible. Please increase the dpi of the image.

3) Section 3.11 Metallic nanoparticles: The authors mention only gold and iron oxide NPs. There is no mention of some of the prominent categories such as; silver NPs, zinc oxide NPs, copper NPs, titanium dioxide NPs, etc. The authors could also broadly term a sub-category as "metal oxide NPs" and include Iron oxide, Zinc Oxide and Titanium dioxide in this section.

At the same time, the authors have only mentioned a couple of studies that utilized Gold and Iron oxide NPs. It would be of readers’ interest to see a more detailed section on these that could include for example a table with the types of metallic NPs, their pros & cons, tumor uptake & specificity in different types of cancers with references. This would provide a deeper understanding to this range of NPs.

4) Tha above comment of including a table with the types of NPs, pros & cons, tumor uptake and specificity also applies to these other sections: solid lipid NPs, genetically-encoded micellar NPs and other polymeric NPs. A tabular illustration with such literature would be an interesting comparison.

The authors nowhere in the manuscript mention the coronal protein adsorption that plays a pivotal role in the obstruction to tumor uptake of NPs. Please add a brief section on this and cite a recent paper Pharmaceutics 2022, 14(1), 41; https://doi.org/10.3390/pharmaceutics14010041

Considering all these above mentioned major revisions, the manuscript would be of readers' interest and useful in the field of nanotechnology.

Author Response

The authors have attempted to do a quick review on the current challenges faced by nanoparticles such as first-pass metabolism and how to solve them. In general, the manuscript requires extensive grammar check and English editing. Additionally, please see some comments below:

1) line 92: Editing required...“Nanoparticles that make it to…”

Our sincere apologies for the typo. We have updated the text.

2) Figure 2: the image resolution is very poor such that the image and text in the image isn’t clearly visible. Please increase the dpi of the image.

We directly downloaded the high resolution image for the figure 2 from the journal website. In the revision we have replaced it with our own illustration to improve resolution.

3) Section 3.11 Metallic nanoparticles: The authors mention only gold and iron oxide NPs. There is no mention of some of the prominent categories such as; silver NPs, zinc oxide NPs, copper NPs, titanium dioxide NPs, etc. The authors could also broadly term a sub-category as "metal oxide NPs" and include Iron oxide, Zinc Oxide and Titanium dioxide in this section.

We thank the reviewer for pointing out this. We have now added silver, zinc oxide, and titanium dioxide nanoparticles.

At the same time, the authors have only mentioned a couple of studies that utilized Gold and Iron oxide NPs. It would be of readers’ interest to see a more detailed section on these that could include for example a table with the types of metallic NPs, their pros & cons, tumor uptake & specificity in different types of cancers with references. This would provide a deeper understanding to this range of NPs.

We thank the reviewer for this valuable suggestion. We have included advantages and disadvantages of various nanoparticle in Table 2 & 3. Additionally, we have also included the mechanism of interaction of the nanoparticles with the biological interface in section 4.

4) Tha above comment of including a table with the types of NPs, pros & cons, tumor uptake and specificity also applies to these other sections: solid lipid NPs, genetically-encoded micellar NPs and other polymeric NPs. A tabular illustration with such literature would be an interesting comparison.

Please refer to our above comment.

The authors nowhere in the manuscript mention the coronal protein adsorption that plays a pivotal role in the obstruction to tumor uptake of NPs. Please add a brief section on this and cite a recent paper Pharmaceutics 202214(1), 41; https://doi.org/10.3390/pharmaceutics14010041

We have included this important issue in section 3.2 and section 4. We have also cited the reference.

Considering all these above mentioned major revisions, the manuscript would be of readers' interest and useful in the field of nanotechnology.

We thank the reviewer for showing interest in our work.

Round 2

Reviewer 1 Report

This is a much improved article following comprehensive revisions - it is clear that the authors have considered and addressed all reviewers' comments appropriately. The article is suitable for publication pending a final proofread to correct minor grammatical errors.

Author Response

We thank the reviewer for the insightful comments and for considering our revision significant.

Reviewer 2 Report

The authors have answered all of the questions posed. The manuscript in its current state is suitable for publishing.

Author Response

We are thankful to the reviewer for the comments and consideration.

Reviewer 3 Report

The manuscript has been thoroughly improved and revised. It is in an acceptable form for publication. However, please cross-check the references once again as I could notice some irregularities at a couple of places.

Author Response

We thank the reviewer for the suggestion. We have made all the necessary changes to the best of our ability.